# Single and Multiplex Immunohistochemistry to Detect Platelets and Neutrophils in Rat and Porcine Tissues

**DOI:** 10.3390/mps5050071

**Published:** 2022-09-19

**Authors:** Stephanie Arnold, Sarah Watts, Emrys Kirkman, Clive P. Page, Simon C. Pitchford

**Affiliations:** 1Sackler Institute of Pulmonary Pharmacology, Institute of Pharmaceutical Science, King’s College London, London SE1 9NH, UK; 2CBR Division, Defence Science and Technology Laboratory, Salisbury SP4 0JQ, UK

**Keywords:** platelets, neutrophils, porcine, rat, immunohistochemistry

## Abstract

Platelet–neutrophil complexes (PNCs) occur during the inflammatory response to trauma and infections, and their interactions enable cell activation that can lead to tissue destruction. The ability to identify the accumulation and tissue localisation of PNCs is necessary to further understand their role in the organs associated with blast-induced shock wave trauma. Relevant experimental lung injury models often utilise pigs and rats, species for which immunohistochemistry protocols to detect platelets and neutrophils have yet to be established. Therefore, monoplex and multiplex immunohistochemistry protocols were established to evaluate the application of 22 commercially available antibodies to detect platelet (nine rat and five pig) and/or neutrophil (four rat and six pig) antigens identified as having potential selectivity for porcine or rat tissue, using lung and liver sections taken from models of polytrauma, including blast lung injury. Of the antibodies evaluated, one antibody was able to detect rat neutrophil elastase (on frozen and formalin-fixed paraffin embedded (FFPE) sections), and one antibody was successful in detecting rat CD61 (frozen sections only); whilst one antibody was able to detect porcine MPO (frozen and FFPE sections) and antibodies, targeting CD42b or CD49b antigens, were able to detect porcine platelets (frozen and FFPE and frozen, respectively). Staining procedures for platelet and neutrophil antigens were also successful in detecting the presence of PNCs in both rat and porcine tissue. We have, therefore, established protocols to allow for the detection of PNCs in lung and liver sections from porcine and rat models of trauma, which we anticipate should be of value to others interested in investigating these cell types in these species.

## 1. Introduction

Animal models of acute lung injury (ALI) and acute respiratory distress syndrome (ARDS) have provided useful insights into the contribution of platelets and neutrophils in the pathogenesis of these diseases, as such models are often investigated in rodents [1,2,3]. One important cause of ALI results from blast exposure, and there is a considerable body of evidence investigating this type of lung injury in large animals, such as pigs, as well as in rodents. Despite experiments being more costly and requiring scientists with highly specialized technical skills, porcine models of trauma allow for accurate laboratory recapitulation of trauma and blast injury, as their physiological similarities and size relative to humans enables direct testing of clinical protocols for the treatment of injury resulting from traumatic injury, including blast injury [4,5]. Blast injuries (resulting e.g., from explosions) are complex injuries, the components of which are characterised by the part of the explosion that caused them [4,5]. Primary blast injury is due to the shock wave, secondary blast injury is due to the fragments and debris energised by the explosion, and tertiary blast injury is due to the rapid movement of the casualty against other objects. Thus, porcine models of blast injury have enabled characterisation of the physiological changes that follow blast exposure, which includes bradycardia, hypotension and changes in respiratory parameters such as apnoea [6,7]. Furthermore, a complex injury model featuring blast injury (primary blast injury) and controlled haemorrhage (simulating the consequences of secondary blast injury), albeit without tertiary blast injury, has been developed and characterised and has already proven advantageous for informing military treatment protocols [8].

Due to lower costs and ease of handling and housing, rodent models have also been used to assess physiological outcomes following blast injury, and the availability of a wide range or reagents and tools has made rodents particularly beneficial for understanding different aspects of the inflammatory response associated with ALI, such as leukocyte recruitment [2,3,9,10], and following blast- and trauma-induced injuries [11,12,13,14].

Immunohistochemistry (IHC) is a very useful and widely used laboratory technique, allowing for the visualisation of in situ proteins in tissues following inflammatory insults [2,15]. However, most of the commercially available reagents used to carry out high quality IHC have been developed for murine, rather than rat experiments, and, of course, for investigating human tissues, including the lung [15]. There are, however, only a very limited number of reagents available for carrying out IHC in other species, such as the pig and rat. Given the usefulness of rats and pigs in investigating the physiological changes accompanying traumatic injury including blast injury, we have, therefore, evaluated a range of reagents that may be of value in these species for IHC of lungs and other tissues. In particular, given the involvement of platelets and neutrophils in lung injury [1,2,3,16], we have investigated a range of reagents for their ability to detect these important cell types immunohistochemically in the lungs and livers of rats and pigs.

## 2. Materials and Methods

### 2.1. Background to In Vivo Experimentation and Tissue Collection

All in vivo experiments reported in this study were carried out in the laboratories of DSTL, Porton Down, Salisbury, UK. All animal experiments were conducted following local ethical approval and in accordance with the Animals (Scientific Procedures) Act 1986 (United Kingdom). These studies were conducted under terminal anaesthesia (animals remained anaesthetised throughout the procedure and were killed humanely at the end of the study with an overdose of anaesthetic). Detailed procedural methodology is provided in previous reports for both the rat trauma and haemorrhagic shock protocol [17], as well as the severe injury and shock with blast exposure model in pigs [7,8,18].

Male rats (6–8 weeks old) were bred from an in-house colony of Porton Wistar rats and had access to chow (LabDiet EURodent Diet 22%) and water ad libitum. The pigs used in this study were female cross-bred large white pigs (43–56 kg), housed indoors, which were fed ad libitum on a complete diet comprising coarse ground mixture of wheat, barley, soya protein, vitamins, and minerals, with free access to water.

### 2.2. Tissue Collection

Rat protocol: six hours after the end of haemorrhage, as described elsewhere [17], all rats were killed humanely with an overdose of anaesthetic (Euthatal, Merial Animal Health Ltd., Harlow, UK). For FFPE lung samples, the accessory lobe was excised and immediately submerged in 4% formaldehyde for a minimum of 24 h. For frozen tissue, representative 1 cm^3^ samples of the left and right lung lobes were flash-frozen in liquid nitrogen.

Pig protocol: pigs were anaesthetised, instrumented, injured, and resuscitated as previously described [18]. All animals received a soft tissue injury and 30% blood volume haemorrhage, but, in addition, a proportion of animals also received a whole-body blast exposure, as has been previously described [7]. Then, 4 h post injury, the pigs were humanely killed with an overdose of sodium pentobarbitone (Euthatal, Merial Animal Health Ltd., Bracknell, UK) given intravenously, and samples of lung and liver were collected post mortem for histological analysis and IHC. For FFPE lung tissue, a representative sample of the accessory lobe was excised and submerged in 4% formaldehyde for a minimum of 24 h, before undergoing standard histoprocessing into FFPE blocks. For frozen lung tissue, a representative 1 cm^3^ piece was sampled from the left caudal lobe and was snap-frozen in liquid nitrogen.

Rat and pig lung tissues were subsequently provided to the Sackler Institute of Pulmonary Pharmacology at King’s College London, where samples were sectioned and IHC protocol optimisation experiments were performed. 

Frozen tissue samples were embedded into OCT and sectioned to 8 micrometres using a cryostat (frozen) or 4 micrometres using a microtome (FFPE) and stored at −80 °C or room temperature, respectively. The negative control was a consecutive tissue section that received all the immunohistochemistry reagents, except for the primary antibody, which enabled detection of unspecific reagent staining. 

### 2.3. Preparation of Rat and Pig Tissue for Immunohistochemistry (IHC)

#### 2.3.1. Frozen Tissue

On the day of IHC, slides were defrosted for 30 min. Sections were fixed for 10 min in 4% formaldehyde diluted in PBS. Endogenous peroxidases were blocked by using 3% hydrogen peroxide (H_2_O_2_) solution. Tissue sections were blocked by pipetting 5% donkey serum in BSA onto tissue sections for 10 min. Donkey serum was removed before adding the primary antibody.

#### 2.3.2. FPPE Tissue

For paraffin-embedded tissue, lung sections were removed of wax by submerging in 100% xylene for 10 min. Sections were rehydrated in decreasing concentrations of ethanol (100%, 90%, 70%) for 1 min in each. Endogenous peroxidases were blocked in the sections by using 3% hydrogen peroxide (H_2_O_2_) solution. Sections then underwent heat-induced epitope retrieval (HIER) for 10 min at 100 degrees Celsius, using a microwave and a domestic pressure cooker. The HIER buffer was microwaved in a pressure cooker until it reached 100 °C, before the slides were added. The following HIER buffers were tested: 1. Tris-EDTA buffer [10 mM Trizma base (Sigma Aldrich, Poole, UK) 1 mM EDTA solution (Sigma Aldrich, Poole, UK), 0.05% Tween 20 (Sigma Aldrich, Poole, UK), pH 9.0]; 2. EDTA buffer [1 mM EDTA (Sigma Aldrich, Poole, UK), pH 8.0]; and 3. sodium citrate buffer [10 mM Sodium citrate (Fisher Scientific, Loughborough, UK), 0.05% Tween 20, pH 6.0]. Once the pressure dropped, the lid of the pressure cooker was removed, and slides were cooled under cold water for 5 min. Tissue sections were blocked by pipetting 5% donkey serum in BSA onto tissue sections for 10 min. Donkey serum was removed before adding the primary antibody. 

Methodologies for IHC (frozen and FFPE): the appropriate primary antibody to detect rat neutrophils (Table 1), rat platelets (Table 2), porcine neutrophils (Table 3), or porcine platelets (Table 4), diluted in 5% donkey serum (in PBS), was added to lung sections and incubated at room temperature for 2 h. For multiplex IHC, primary antibodies were diluted together in 5% donkey serum in Tris-buffered saline (TBS), and the incubation step was for 1 h. Primary antibodies were washed off using PBS (or TBS for multiplex protocol). Excess PBS (or TBS) was removed, and the appropriate biotinylated secondary antibody was added (Table 5). Secondary antibody was incubated on sections for 1 h at room temperature. Secondary antibody was washed in PBS (or TBS) under gentle agitation for 5 min.

### 2.4. Rat Monoplex Horse Radish Peroxidase Detection and Colour Development

The Vectastain Elite ABC horseradish peroxidase (HRP) detection kit (PK-6100, Vector Laboratories, Burlingame, CA, USA) was prepared 30 min prior to use, in accordance with the instructions of the manufacturer (0.5% solution A and 0.5% solution B diluted in 5% donkey serum). Following washing, excess PBS was removed from slides, and the Vectastain Elite ABC HRP detection kit was pipetted onto each tissue section for 30 min. The Vectastain Elite ABC HRP reagent was then washed off in PBS under gentle agitation for 5 min. Sections were submerged in 3, 3′-diaminobenzidine (DAB) development buffer (0.1 M Trizma base (pH 7.6), 5% 3, 3′-DAB and 0.3% H_2_O_2_) for 10 min. DAB forms a brown precipitate in regions of positive antigen detection.

### 2.5. Rat Multiplex Alkaline Phosphatase and Horse Radish Peroxidase Detection and Colour Development

ImmPRESS-AP Anti-Rabbit IgG (alkaline phosphatase) polymer detection kit (MP-5401, Vector Laboratories, Burlingame, CA, USA) was used at the recommended working concentration, in accordance with the instructions of the manufacturer. Following washing, excess TBS was removed and ImmPRESS-AP Anti-Rabbit IgG (alkaline phosphatase) polymer detection reagent was added to each tissue section. The AP reagent was incubated on tissue sections for 30 min at room temperature before being washed off using TBS under gentle agitation for 5 min.

The Vectastain Elite ABC horseradish peroxidase (HRP) detection kit (PK-6100, Vector Laboratories, Burlingame, CA, USA) was prepared 30 min prior to use, in accordance with the instructions of the manufacturer (0.5% solution A and 0.5% solution B diluted in 5% donkey serum). Following washing, excess TBS was removed and 100 μL of the Vectastain Elite ABC HRP detection kit was pipetted onto each tissue section for 30 min at room temperature. The Vectastain Elite ABC HRP reagent was washed off using TBS before being submerged in TBS under gentle agitation for 5 min.

For alkaline phosphatase dependent detection, the Vector Blue alkaline phosphatase (AP) substrate kit (SK-5300, Vector Laboratories, Burlingame, CA, USA) was used. This was prepared in accordance with the instructions of the manufacturer, with the addition of levamisole (SP-5000, Vector Laboratories, Burlingame, CA, USA), an endogenous AP inhibitor, that was added in accordance with the instructions of the manufacturer. Following washing, excess TBS was removed and Vector Blue AP colour substrate was added to each tissue section for 10 min at room temperature. Vector Blue AP colour substrate was washed off in TBS under gentle agitation for 5 min. For HRP dependent detection, the NovaRED HRP substrate kit (SK-4800, Vector Laboratories, Burlingame, CA, USA) was used. This was prepared in accordance with the instructions of the manufacturer. Following washing, excess TBS was removed and NovaRED HRP substrate was added to each tissue section for 20 min at room temperature. The substrate was washed off in TBS under gentle agitation for 5 min.

In experiments where both primary antibodies were raised in the same species, a sequential staining protocol was used, where an antibody denaturation step ‘antigen stripping’ was used before the application of the 2nd primary antibody [19]. Thus, following Vector Blue AP substrate development, sections underwent a second heat-induced epitope retrieval (HIER) for antibody denaturation: Tris-EDTA buffer (10 mM Trizma base (Sigma Aldrich, Poole, UK) 1 mM EDTA solution (Sigma Aldrich, Poole, UK), 0.05% Tween 20 (Sigma Aldrich, Poole, UK), pH 9.0) for 5 min at 100 °C, by being microwaved in a pressure cooker. Once the pressure dropped, the lid of the pressure cooker was removed, and slides were cooled under cold running water for 5 min. It was noted that the first chromogen needed to be Vector Blue because the NovaRED substrate did not withstand the denaturation process. 

A schematic diagram shows the distinctions between the monoplex and multiplex procedures (Figure 1).

### 2.6. Counterstaining, Dehydration and Clearing

Sections were submerged into Hematoxylin Solution, Gill No. 1, for 2 min before being washed in tap water (monoplex only). Sections were briefly submerged in differentiation solution for 3 s, before being washed for a final time in tap water. Sections were dehydrated in increasing concentrations of ethanol (70%, 90% and 100%) by being submerged for 1 min in each ethanol concentration (monoplex), or 10 s in each ethanol concentration, to avoid solubilising colour substrates (multiplex). Sections were cleared in 100% xylene for 10 min and cover-slipped with DPX mounting medium (monoplex) or cleared with Histoclear for 10 min and cover-slipped with Vectamount, a non-xylene based mounting medium (multiplex) used in histology. Images were captured using an LED upright bright-field microscope (DM2000 Leica, Nussloch, Germany) attached to a bright-field microscope camera (DCF295, Nussloch, Germany). 

## 3. Results

### 3.1. Protocol Optimisation for IHC in Rat Tissues to Identify Neutrophils and Platelets Using Singleplex Protocols

Protocol optimisation experiments were performed to identify a suitable methodology for IHC to detect platelets and neutrophils in rat tissues. Due to the limited number of reagents available that are specific for the rat, cross-reactivity with antibodies that have been made to target human and mouse antigens were tested. Both formalin-fixed paraffin-wax-embedded (FFPE) and frozen tissues were also tested, with different antigen retrieval buffers tested for the FFPE tissues. Antibodies were tested on FFPE samples initially, since this is the preferred method for optimal preservation of tissue morphology. When antibodies failed in FFPE tissue, for example, all antibodies tested for rat platelets, we tested antibodies in frozen tissues. For the PNC staining, it was essential that both antibodies were suitable for the same sample-preparation type. Several concentrations of each antibody were tested, determined through the published literature and user reviews on the manufacturers’ websites. 

Several primary antibodies were tested to detect neutrophils in rat tissues (Table 1). The only successful neutrophil-specific antibody out of those tested was an anti-myeloperoxidase (MPO) antibody (ab9535, Abcam, Cambridge, UK). This anti-MPO antibody was detected using an anti-rabbit biotinylated secondary antibody (BA1000, Vector Laboratories, Burlingame, CA, USA), Vectastain Elite ABC HRP detection kit (PK-6100, Vector Laboratories, Burlingame, CA, USA) and DAB development buffer (Sigma Aldrich, Poole, UK). The antibody worked effectively in both paraffin-wax-embedded and frozen-tissue sections (Figure 2A). No unspecific binding of detection reagents was observed in a consecutive section that received no primary antibody, indicating no unspecific binding of detection reagents (Figure 2B). A haematoxylin nuclear counterstain revealed that positively stained regions were associated with cells featuring a multi-lobed nucleus, a key characteristic of neutrophils (Figure 2C). Neutrophils could be detected proximal to blood vessels in the lung (Figure 2D). As MPO is an enzyme secreted from neutrophils upon degranulation, extracellular MPO positivity was also observed, particularly in the liver and within focal necrotic regions (Figure 2E,F).

Primary antibodies were also tested to detect platelets in rat tissues (Table 2). Antibodies were selected based on those already successfully used by the Sackler Institute of Pulmonary Pharmacology at King’s College London, in the published literature and per user reviews on supplier websites. The only successful platelet-specific antibody out of those tested was an anti-CD61 antibody (554951, BD Biosciences, Franklin Lakes, NJ, USA), but this only worked when used with frozen tissue. This anti-CD61 antibody was detected using an anti-rabbit biotinylated secondary antibody (BA1000, Vector Laboratories, Burlingame, CA, USA), Vectastain Elite ABC horseradish peroxidase (HRP) detection kit (PK-6100, Vector Laboratories, Burlingame, CA, USA) and DAB development buffer. Platelets could be observed within blood vessels and within the lung parenchyma (Figure 3A). No unspecific binding of detection reagents was observed in a consecutive section that received all detection reagents with no primary antibody (Figure 3B). Platelets could be observed within both lung tissues (Figure 3C,D) and liver tissues (Figure 3E,F).

Protocol optimisation for immunohistochemistry in rat tissues identified PNCs using a multiplex protocol. As the only suitable platelet specific antibody worked solely in frozen-tissue sections, the detection of PNCs was limited to analysis of tissue sections in this format. To detect PNCs, both primary platelet and neutrophil antibodies (anti-MPO and anti-CD61) were incubated together on the same section. As each antibody was raised in a different species, different detection systems could be used simultaneously to detect the primary antibodies. Primary antibodies were, therefore, detected using a HRP-dependant reaction and an AP-dependant reaction. As PNCs are associated with colocalisation of both cell types, blue and red detection systems were chosen to detect any overlapping events that may not be visible using DAB substrate (Table 6). PNC staining was observed in both the rat lung (Figure 4A,B) and liver tissues (Figure 4C,D).

### 3.2. Protocol Optimisation for IHC to Identify Neutrophils and Platelets in Pig Tissues

Protocol optimisation experiments were performed to identify a suitable methodology for IHC to detect platelets and neutrophils in pig tissues. Several antibodies were tested, based on the literature reviews and user reviews on supplier websites (Table 3). The only successful neutrophil specific antibody out of those tested was an anti-MPO antibody (ab9535, Abcam, Cambridge, UK), the same antibody we had identified as successful for IHC in rat tissues, as described above. The antibody worked effectively in both FFPE and frozen issue; as pig platelets could also be detected in FFPE, and this method optimally preserves tissue morphology, we focused on the FFPE neutrophil staining (Figure 5A). No unspecific binding of detection reagents was observed in a consecutive section that received no primary antibody (Figure 5B). Like for the rat tissue, neutrophils could be visualised within blood vessels in the lung (Figure 5C,D) and were localised to focal necrotic regions in the liver (Figure 5E,F).

Several primary antibodies were tested to detect platelets in pig tissues (Table 4). Antibodies were selected based on the published literature and user reviews on supplier websites. Out of those tested, two antibodies showed positive platelet staining, which were an anti-CD42b antibody (554951, BD Biosciences, Franklin Lakes, NJ, USA) and an anti-CD49b antibody (ab181548, Abcam, Cambridge, UK). The anti-CD42b antibody worked in both FFPE tissue and frozen tissue, whilst the anti-CD49b antibody only worked in frozen tissue (Figure 6 and Figure 7). Platelets were observed within pulmonary blood vessels (Figure 6A and Figure 7A). No unspecific binding of detection reagents was observed in consecutive sections (Figure 6B and Figure 7B). Platelets were identified with both antibodies in pulmonary blood vessels (Figure 6C), which were not present in the negative reagent control (Figure 6D), as well as multi-lobular nucleated cells (Figure 7C), plus there were platelet aggregates in regions of alveolar haemorrhage (Figure 7D). Platelets were detected in the liver with both antibodies (Figure 6E,F and Figure 7E,F).

### 3.3. Protocol Optimisation for IHC in Porcine Tissues to Identify PNCs Using a Multiplex Protocol

Platelet and neutrophil antibodies were effective in both FFPE and frozen tissue. However, since FFPE tissue preparation retains better tissue morphology than frozen tissue, PNCs were detected in FFPE tissue. However, since both platelet (anti-CD42b) and neutrophil (anti-MPO) antibodies were both raised in rabbits, a method was developed based on the previous literature [20]. Instead of adding primary antibodies together, the neutrophil (anti-MPO) primary antibody, AP detection reagents and AP colour substrate (blue) were added, before an additional HIER step to denature the residual antibody and reagents. This step was followed by addition of the platelet (anti-CD42b) antibody, HRP detection reagents and HRP colour substrate (red). It was critical to use the colour substrates in this order, because the red substrate is not heat stable. PNCs are associated with colocalisation of both cell types, so blue and red detection systems were chosen to detect any overlapping events that may not be visible with DAB substrate. Lung CD42b can be observed in red (Figure 8A), and lung MPO can be observed in blue (Figure 8B). When multiplexed, colocalised events could be observed in the lung (Figure 8C,D) and liver (Figure 8E,F). 

## 4. Discussion

Advances in trauma care, particularly with regard to haemorrhage control and resuscitation, have resulted in increased survival for the severely injured. However, many survivors develop multi-organ dysfunction syndrome (MODS) and failure due to the systemic response to injury and the development of SIRS (systemic inflammatory response syndrome). There is, therefore, an interest in better understanding MODS after severe trauma [21,22,23]. It is important to understand the origins of MODS, including the inflammatory mechanisms contributing to MODS, which in part requires the assessment of tissues from animals following traumatic injury, including blast injury. However, the availability of reagents to investigate inflammatory pathways in the animal species used for modelling blast injury, namely the rat and pig, is poor, with most reagents being developed for use in mice or humans. In the present study, we have, therefore, carried out an extensive search and evaluation of commercially available reagents to detect platelets and neutrophils in tissues using IHC, to determine whether any of these could be suitable to evaluate inflammatory changes in rats and pigs that have been exposed to blast injury, which could be used in the future in these species to better understand the inflammatory chances associated with MODS.

To determine inflammatory cell recruitment into lung and liver in rat and pig models of traumatic haemorrhagic shock, with and without thoracic blast injury, we have evaluated various anti-human and anti-mouse antibodies for cross-reactivity, which have been reported to target platelets and neutrophils in our laboratory and others [2,24,25,26,27]. Furthermore, we have investigated the use of these antibodies to detect PNCs as the published literature describing histological protocols for detecting platelets and neutrophils in porcine and rodent tissues is limited.

An anti-human MPO antibody yielded successful and specific neutrophil staining of tissues from both species, using frozen and FFPE tissues. Although MPO is a neutrophil activation marker, positive staining was mostly observed intracellularly, suggesting it was of value in enumerating neutrophils in tissues. For the detection of platelets in pig tissues, an antibody recognising CD49b worked well in frozen tissue, whilst CD42b worked in both frozen and FFPE tissues. This anti-CD42b antibody yielded similar staining results in the FFPE pig tissue to those observed in FFPE murine lung tissue in a recent study by our laboratory [2]. We have previously used a CD42b antibody for platelet quantification because 1) FFPE-fixed tissues retain better morphology, and 2) FFPE tissue fixation and processing techniques also allow for retention of the RBCs in the tissues [28]. This is of interest, given that alveolar haemorrhage is a major feature of blast injury, and we have previously reported that red blood cells (and their products) can induce an inflammatory response in the lung, which is dependent on platelets [28]. However, surprisingly, anti-mouse antibodies recognising CD42b and CD49b did not adequately stain platelets in rat tissues, so we have still to identify a suitable platelet antibody for use in FFPE rat tissue. However, during the protocol optimisation period, another laboratory reported rat platelet staining using an anti-CD61 antibody for immunofluorescence microscopy [29]. Whilst this antibody was also successful in staining for platelets in frozen rat tissue in a blast-injury model, it was not suitable for use in FFPE tissue, despite testing several different antigen retrieval conditions.

Importantly, given that platelets and neutrophils are known to co-operate during inflammatory processes and often form PNCs, we could also quantify the presence of PNCs in tissues using the same antibodies that were used for single-stain IHC. We were successful in the development of a multiplex IHC protocol that enabled dual colour detection of double-positive, colocalised events. 

Whilst chromogenic IHC facilitates investigation into cellular recruitment and resolves spatial information of cells and their colocalisation, there are some limitations of the technique. Whilst cells can be enumerated manually or be measured digitally using image analysis (IA) software, chromogenic IHC is a semiquantitative method due to the signal amplification steps involved. An amplification of signal by IHC is also a limitation of quantifying cellular colocalisation, as this may cause proximity of proteins to be exaggerated, thus creating false positive events. An alternative method that may be useful to observe colocalised events such as PNCs is a proximity ligation assay (PLA), as, using this methodology, a signal is only observed when two proteins of interest are expressed at the same cellular location [30]. In addition, multiplex chromogenic IHC is also limited by overlapping colours in the brightfield light spectra, which can be overcome by using multiplex immunofluorescence (mIF). Advances in mIF have enabled analysis of up to eight biomarkers in just one tissue section, which is achievable using both automated and manual staining methods [31]. Furthermore, emerging technologies are making it possible to measure up to 100 markers in one tissue section, with the capability to overlay both RNA and protein expression [32].

In conclusion, the methods described here will help characterise the nature of platelet–neutrophil interactions in species used to model the pathology of blast-trauma-related injury, a heterotypical interaction that may have multiple consequences and temporal distinctions across different body compartments [2,33,34], as well as being of value for the investigation of platelets and neutrophils in other experimental models in rats and pigs. 

## Figures and Tables

**Figure 1 mps-05-00071-f001:**
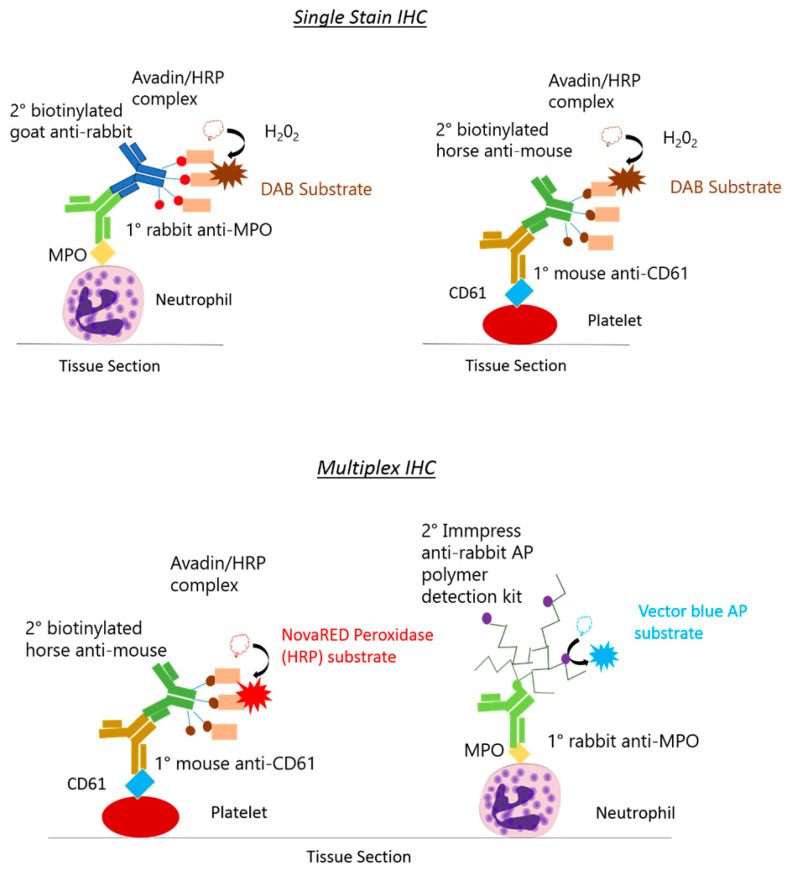
**Schematic of single-stain and multiplex immunohistochemistry reporter systems.** In single-stained immunohistochemistry (IHC) (above), neutrophils and platelets are detected in separate tissue sections using a horse radish peroxidase (HRP)-dependent reporting system. The 3,3’-diaminobenzidine (DAB) substrate forms a brown precipitate in positively stained regions. In multiplex IHC (below), platelets and neutrophils can be detected in the same tissue section. Platelets are detected using a HRP-dependent reaction, whilst neutrophils are detected using an alkaline phosphatase (AP) dependent reaction. Blue and red colour substrates allow for visualisation of colocalised events.

**Figure 2 mps-05-00071-f002:**
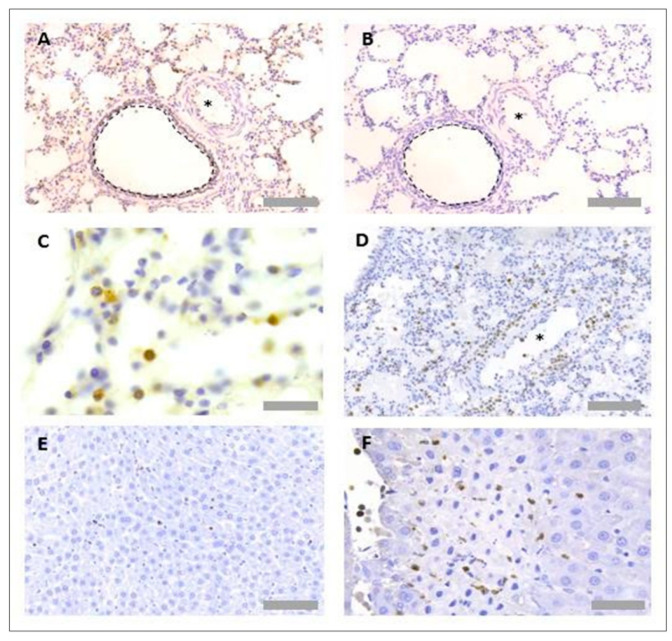
**Neutrophil immunohistochemistry in rat lung and liver tissues.** For protocol optimisation, sections were immunostained with primary antibody (**A**) and without primary antibody (**B**), to detect any unspecific binding of detection reagents. Representative images of neutrophil immunohistochemistry (anti-MPO) in FFPE rat lung tissue (**C**,**D**) and FFPE rat liver tissue (**E**,**F**) are also shown. Asterisks denote blood vessels; black dotted lines denote airway wall. Scale bar = 100 µm in images (**A**,**B**,**D**,**E**) (×20 objective), 50 µm in image (**F**) (×40 objective) and 20 µm in image (**C**) (×63 objective).

**Figure 3 mps-05-00071-f003:**
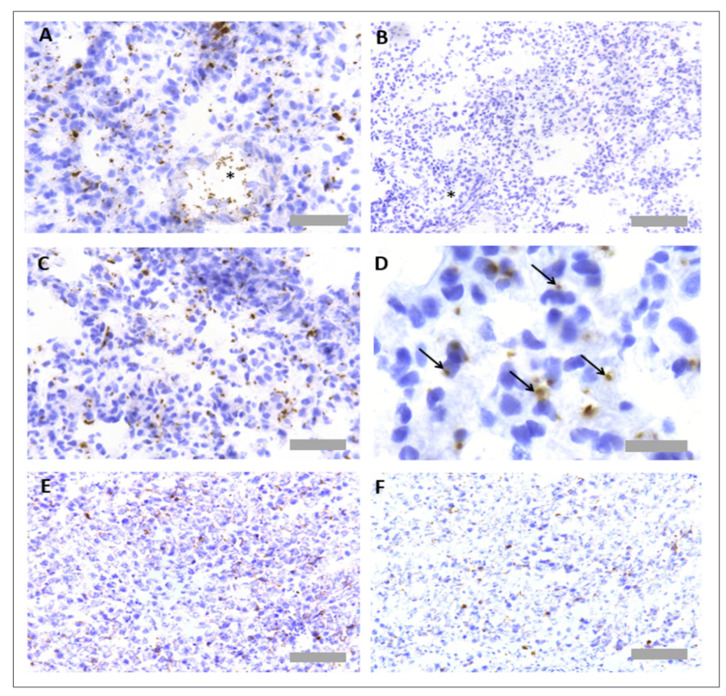
**Platelet immunohistochemistry in rat lung and liver tissues.** For protocol optimisation, sections were immunostained with primary antibody (**A**) and without primary antibody (**B**), to ensure no unspecific binding of detection reagents. Representative images of platelet immunohistochemistry (anti-CD61) in frozen rat lung tissue (**C**,**D**) and frozen rat liver tissue (**E**,**F**) are also shown. Asterisks denote blood vessels. Arrows denote CD61+ events. Scale bar = 50 µm in (**A**,**C**) (×40 objective), 100 µm in (**B**,**D**,**E**,**F**) (×20 objective).

**Figure 4 mps-05-00071-f004:**
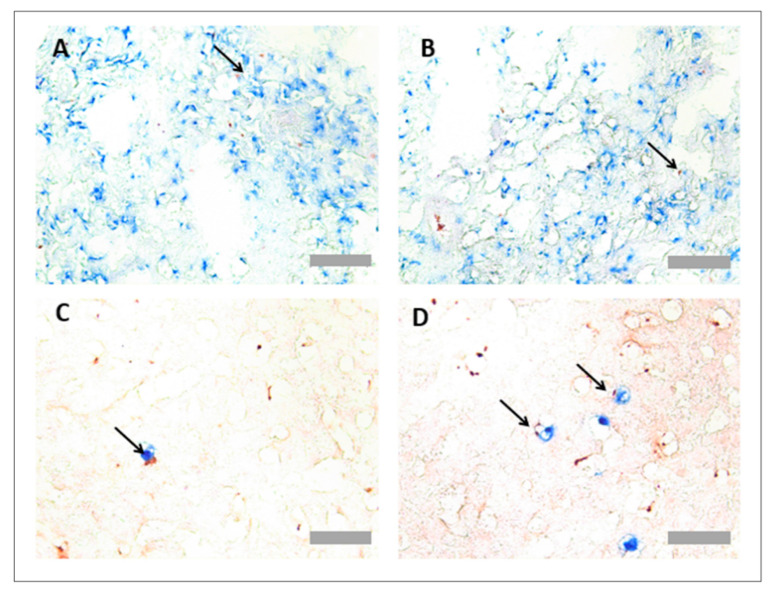
**Platelet neutrophil complex multiplex immunohistochemistry in rat lung and liver tissues.** Frozen rat tissues were prepared and immunostained with anti-myeloperoxidase (MPO) antibody (ab9535, Abcam, UK) and anti-CD61 antibody (PB9647, Boster, USA), prepared at 1/25 and 1/100 dilutions, respectively. Platelets appear red, and neutrophils appear blue. Representative images of platelet neutrophil complex (PNC) staining in rat lung tissue (**A**,**B**) and rat liver tissue (**C**,**D**). Arrows denote CD61+ MPO+ events. Scale bar = 100 µm in Figures (**A**,**B**) (×20 objective) and 30 µm in Figures (**B**,**C**) (×63 objective).

**Figure 5 mps-05-00071-f005:**
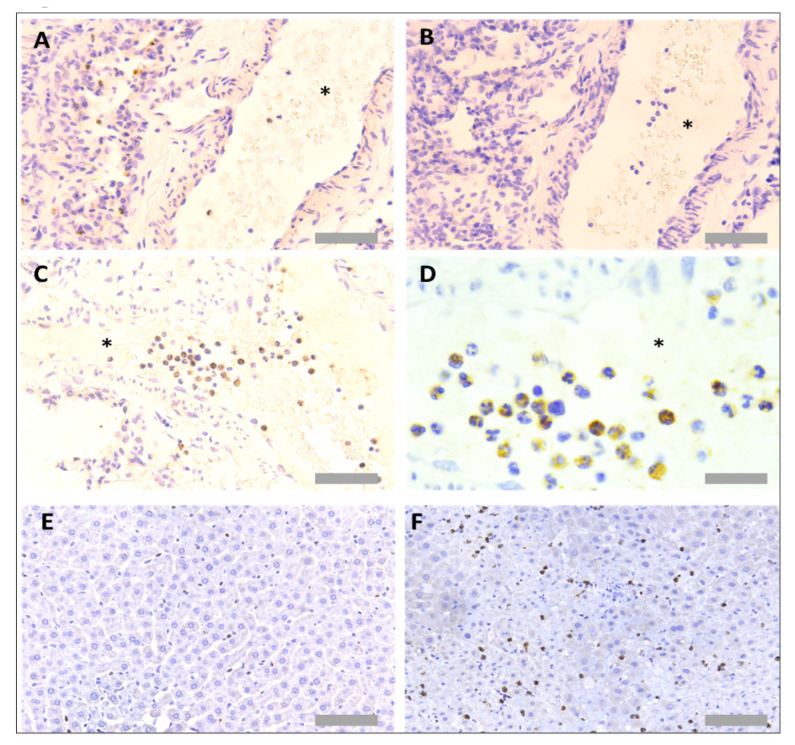
**Neutrophil immunohistochemistry in pig lung and liver tissues.** For protocol optimisation, consecutive sections were immunostained with primary antibody (**A**) and without primary antibody (**B**), to ensure no unspecific binding of detection reagents. Representative images of neutrophil immunohistochemistry (anti-MPO) in FFPE pig lung tissue (**C**,**D**) and FFPE pig liver tissue (**E**,**F**) are also shown. Asterisks denote blood vessels. Scale bar = 100 µm in Figures (**A**–**C**,**E**,**F**) (×20 objective), 20 µm in Figure (**D**) (×63 objective).

**Figure 6 mps-05-00071-f006:**
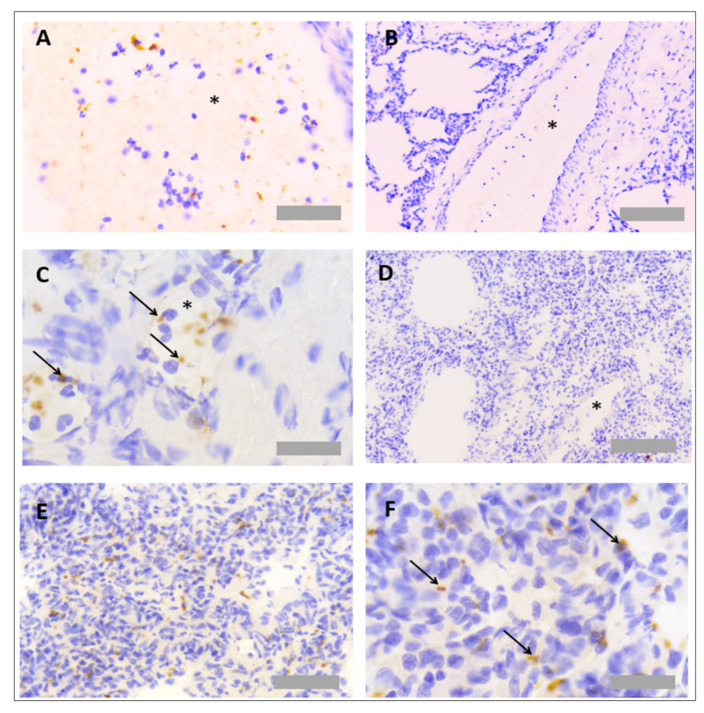
**Platelet immunohistochemistry in frozen pig lung and liver tissues.** Consecutive sections were immunostained with primary antibody (**A**) and without primary antibody (**B**), to detect unspecific binding of detection reagents. Representative images of platelet immunohistochemistry (anti-CD49b) in frozen pig lung tissue (**C**) are also shown. Signal is absent in the reagent only control (**D**). Platelets were also observed in the liver tissue (**E**,**F**). Asterisks denote blood vessels. Arrows denote CD49+ platelet events. Scale bar = 50 µm in Figures (**A**,**E**) (×40 objective), 100 µm in Figures (**B**,**D**) (×20 objective) and 20 µm in Figures (**C**,**F**) (×63 objective).

**Figure 7 mps-05-00071-f007:**
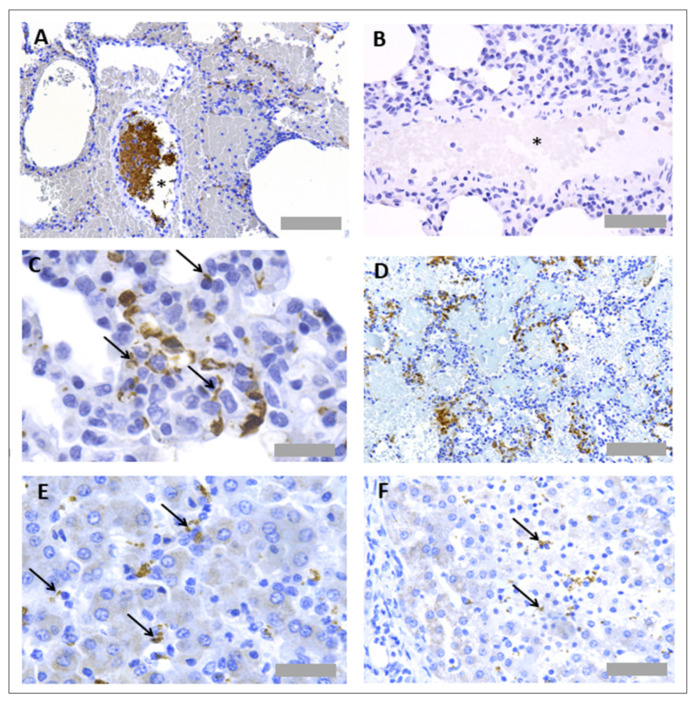
**Platelet immunohistochemistry in FFPE pig lung and liver tissues**. Sections were immunostained with primary antibody (**A**) and without primary antibody (**B**), to ensure no unspecific binding of detection reagents. Representative images of neutrophil immunohistochemistry (anti-CD42b) in FFPE pig lung tissue (**C**,**D**) and FFPE pig liver tissue (**E**,**F**) are also shown. Asterisks denote blood vessels. Arrows denote CD42b+ platelet events. Scale bar = 100 µm in Figures (**A**,**D**) (×20 objective), 50 µm in Figure (**B**,**F**) and 30 µm in Figures (**C**,**E**) (×63 objective).

**Figure 8 mps-05-00071-f008:**
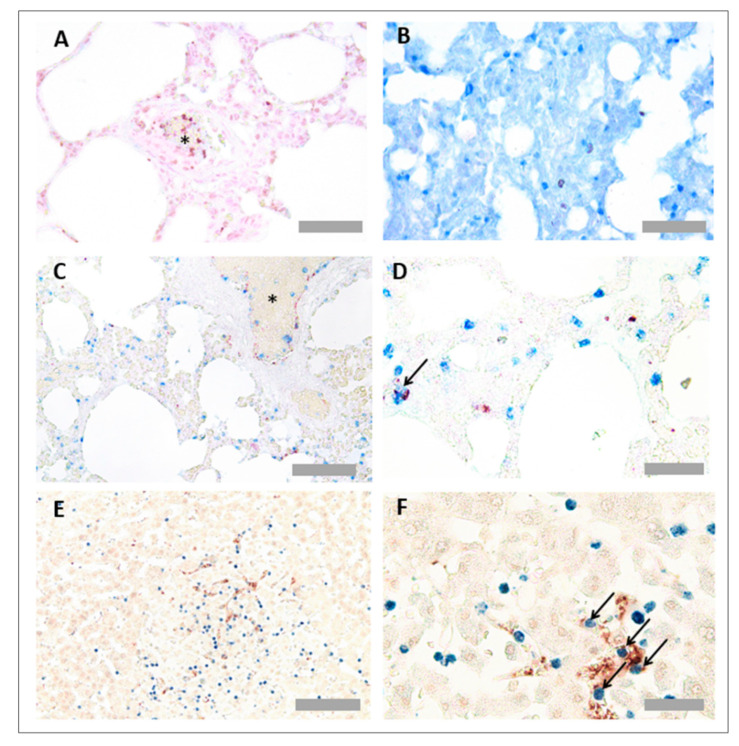
**Platelet neutrophil complex multiplex immunohistochemistry in pig lung and liver tissues.** To ensure colour detection systems were functional, separate sections were immunostained with either anti-CD42b shown in red (**A**) or anti-MPO shown in blue. Representative images of neutrophil immunohistochemistry (anti-CD42b) in pig lung tissue (**C**,**D**) and pig liver tissue (**E**,**F**) are also shown. Asterisks denote blood vessels. Arrows point to PNCs. Scale bar = 50 µm in Figures (**A**,**B**) (×40 objective), 100 µm in Figure (**C**,**E**) (×20 objective) and 30 µm in Figures (**C**,**E**) (×63 objective) for neutrophil complex multiplex immunohistochemistry in pig lung and liver tissues. Formalin fixed paraffin wax embedded (FFPE), and pig tissues were prepared and immunostained with anti-myeloperoxidase (MPO) antibody (ab9535, Abcam, Cambridge, UK) and anti-CD42b antibody (ab183345, Abcam, Cambridge, UK), prepared at 1/50 and 1/100 dilutions, respectively. Platelets appear red, and neutrophils appear blue. To ensure colour detection systems were functional, separate sections were immunostained with either anti-CD42b shown in red (**A**) or anti-MPO shown in blue. Representative images of neutrophil immunohistochemistry (anti-CD42b) in pig lung tissue (**C**,**D**) and pig liver tissue (**E**,**F**) are also shown. Asterisks denote blood vessels. Arrows point to PNCs. Scale bar = 50 µm in Figures (**A**,**B**) (×40 objective), 100 µm in Figure (**C**,**E**) (×20 objective) and 30 µm in Figures (**C**,**E**) (×63 objective).

**Table 1 mps-05-00071-t001:** Neutrophil-specific antibodies tested in rat tissue.

Target Antigen	Supplier	Catalogue Number	Antibody Raised In	Dilutions Tested	Success	Optimum HIER Buffer
Ly-6G	Bioss, USA	bs-2576R	rabbit	1/100, 1/500	×	N/A
MPO	Abcam, UK	ab9535	rabbit	1/25 1/50, 1/100 1/500	1/50 in frozen and FFPE	EDTA pH 8
MPO	Abcam, UK	ab90810	mouse	1/50, 1/100 1/500	×	N/A
Neutrophil Elastase	Abcam, UK	ab68672	rabbit	1/50, 1/100 1/200, 1/500	×	N/A

**Table 2 mps-05-00071-t002:** Platelet-specific antibodies tested in rat tissue.

Target Antigen	Supplier	Catalogue Number	Species Raised In	Dilutions Tested	Success
CD41	Santa Cruz Biotechnologies, USA	sc-6602	goat	1/100 1/500	×
CD41	Santa Cruz Biotechnologies, USA	sc-6604	goat	1/100 1/500	×
CD41	Abcam, UK	ab93983	rabbit	1/50 1/100 1/500	×
CD42b	Abcam, UK	ab183345	rabbit	1/50 1/100 1/200 1/500	×
CD42c	Santa Cruz Biotechnologies, USA.	sc-7073	goat	1/50 1/100 1/500	×
CD49b	Abcam, UK	ab181548	rabbit	1/100 1/200 1/500	×
CD61	BD Biosciences, USA	MCA2263GA	mouse	1/50 1/100 1/500	×
CD61	Boster, USA	PB9647	rabbit	1/50 1/100 1/500	×
CD61	BD Biosciences, USA	554951	mouse	1/50 1/100 1/500	1/50 in frozen tissue

**Table 3 mps-05-00071-t003:** Neutrophil-specific antibodies tested in pig tissue.

Target Antigen	Supplier	Catalogue Number	Antibody Raised In	Dilutions Tested	Success	Optimum HIER Buffer
Ly-6G	BioXCell, USA	BE00751	rat	1/100 1/1000	×	N/A
Ly-6G	Bioss, USA	bs-2576R	rabbit	1/100 1/500	×	N/A
MPO	Abcam, UK	ab9535	rabbit	1/25 1/50 1/100 1/500	1/50 in frozen and FFPE	EDTA pH 8
MPO	Abcam, UK	ab90810	mouse	1/50, 1/100 1/500	×	N/A
Neutrophil (NIMP-R14)	Abcam, UK	ab2557	rat	1/100 1/500	×	N/A
PSGL-1	Abcam, UK	ab110096	rat	1/50, 1/500	×	N/A

**Table 4 mps-05-00071-t004:** Platelet-specific antibodies tested in pig tissue.

Target Antigen	Supplier	Catalogue Number	Species Raised In	Dilutions Tested	Success	Optimum HIER Buffer
CD41	Santa Cruz Biotechnologies, USA	sc-6602	goat	1/100 1/500	×	N/A
CD41	Santa Cruz Biotechnologies, USA	sc-6604	goat	1/100 1/500	×	N/A
CD41	Abcam, UK	ab33661	rat	1/100 1/500	×	N/A
CD41	Abcam, UK	ab93983	rabbit	1/50, 1/100 1/500	×	N/A
CD42b	Abcam, UK	ab183345	rabbit	1/50, 1/100 1/200 1/500	1/100 FFPE and Frozen	Tris-EDTA pH 9.0
CD42c	Santa Cruz Biotechnologies, USA	sc-7073	goat	1/50, 1/100 1/500	×	N/A
CD49b	Abcam, UK	ab181548	rabbit	1/100, 1/200 1/500	1/100 FFPE	Tris-EDTA pH 9.0
CD61	BD Biosciences, USA	JM2E5	mouse	1/50, 1/100 1/500	×	N/A

**Table 5 mps-05-00071-t005:** Secondary antibody reagents for rat and pig tissues.

Target Species	Supplier	Catalogue Number	Antibody Raised In	Dilutions Tested
**Goat**	Santa Cruz, USA	sc-3854	donkey	1/100, 1/200
**Mouse**	Vector, USA	BA2001	horse	1/200
**Rabbit**	Vector, USA	BA1000	horse	1/200

**Table 6 mps-05-00071-t006:** Staining strategy for co-localisation events.

Cell Type	Fixation Method	Primary Antibody	Secondary Antibody	Tertiary Reagent	Colour Substrate
**Neutrophils**	Frozen	Rabbit anti-human myeloperoxidase (MPO) ab9535 (Abcam, UK) Concentration: 1/25 (1 h)	ImmPRESS-AP Anti-Rabbit IgG (alkaline phosphatase) Polymer Detection Kit MP-5401 (Vector Laboratories, USA) (30 min)	N/A	Vector Blue Alkaline Phosphatase (AP) substrate kit SK-5300 (Vector Laboratories, USA) (10 min)
**Platelets**	Frozen	Mouse anti-rat CD61 PB9647 (Boster, USA) Concentration: 1/100 (1 h)	Biotinylated horse anti-mouse (BA-2001), (Vector Laboratories, USA) Concentration: 1/200 (1 h)	Avadin and biotinylated HRP enzymes VECTASTAIN elite ABC kit PK-6100, Vector Laboratories, USA (30 min)	NovaRED Peroxidase (HRP) substrate kit SK-4800 (Vector Laboratories, USA) (20 min)

## Data Availability

The data presented in this study are available on request from the corresponding author.

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
