# Peer review of "Single and Multiplex Immunohistochemistry to Detect Platelets and Neutrophils in Rat and Porcine Tissues"

_mps, 2022, doi:10.3390/mps5050071_

Round 1
Reviewer 1 Report
In this study, the authors investigated various types of reagents capable of immunohistochemically detecting the cell types - platelets and neutrophils present in lung injury - in pigs and rats. I agree that the immunohistochemistry is widely used in laboratories allowing visualization of in situ proteins in tissues.
From the methodological point of view, the proposal seems original, and I think that can help in research. However, there are many variables such as reagents, antibodies, in addition to the detection kits. I think protocols should be clearer regarding reagents and samples used. For example, were all reagents used only on frozen samples? In this title “Preparation of rat and pig tissue for immunohistochemistry (IHC)”, it’s clear that the authors prepared the frozen tissue for IHC. Was the same protocol used for FFPE?
I also have some doubts about the use of formalin in tissue. Why the authors used 4% formalin? Generally, 10% buffered formalin or 4% paraformaldehyde is used.
I have some considerations about the Tables. I think Tables should have titles for better understanding. And regarding the abstract, I suggest writing in full the numbers, as one (not 1), two (not 2).
I strongly suggest a review of methods and protocols to clarify the differences between reagents, antibodies and to explain whether the study was performed only on frozen material or on both, frozen and paraffin samples of tissues.
It is interesting that the methods and protocols are consistent with the results obtained.
Reviewer 2 Report
In this manuscript the authors evaluated commercially available antibodies to detect platelet and/or neutrophil in rat or pig lung and liver tissues
embedded in paraffin o frozen. The visualization of proteins was be by immunohistochemistry used HRP or alkaline phosphatase detection kit.
The description of methods is very detail but I encourage the authors to address few concerns to further strengthen for acceptation in the journal: In general, should include a good description of the figures
e.g. Page 8 Line 225 “No unspecific binding of detection reagents was observed in a consecutive section that received no primary antibody, indicating no unspecific binding of detection reagents (Figure 2)”. In the Figure 2 are six panels, What panel is described in this sentence?
Reviewer each figure and describe all panels includes, in the figures 2 and 5 incorporate in the image the arrows and explain that is observed
